# Unified Multi-Abstraction-Level Functional Testing and Performance Measurements for Industrial IoT in Spatially Distributed Narrow Band-Wireless Wide Area Networks

**DOI:** 10.3390/s24237579

**Published:** 2024-11-27

**Authors:** Jubin Sebastian E, Fabian Sowieja, Axel Sikora

**Affiliations:** Institute of Reliable Embedded Systems and Communication Electronics (ivESK), Offenburg University of Applied Sciences, 77652 Offenburg, Germany; fabian.sowieja@hs-offenburg.de (F.S.); axel.sikora@hs-offenburg.de (A.S.)

**Keywords:** IoT, SDWN, LPWAN, test and measurements, test methodology, testbeds

## Abstract

Narrow Band-Wireless Wide Area Networking (NB-WWAN) technologies are becoming more popular across a wide range of application domains due to their ability to provide spatially distributed and reliable wireless connectivity in addition to offering low data rates, low bandwidth, long-range, and long battery life. For functional testing and performance assessments, the wide range of wireless technology alternatives within this category poses several difficulties. At the device level, it is necessary to address issues such as resource limitations, complex protocols, interoperability, and reliability, while at the network level, challenges include complex topologies and wireless channel/signal propagation problems. Testing the functionality and measuring the performance of spatially distributed NB-WWAN systems require a systematic approach to overcome these challenges. Furthermore, to provide a seamless test flow, it is also critical to test and compare the performance of wireless systems systematically and consistently across the different system development phases. To evaluate NB-WWAN technologies comprehensively across multiple abstraction levels—network simulators, emulated lab testbeds, and field test environments—we propose a unified multi-abstraction-level testing methodology. A detailed technical description of the prototype implementation and its evaluation is presented in this paper.

## 1. Introduction

Many application domains have been drawn to Spatially Distributed Wireless Networking (SDWN) technologies as a result of the digitalization and wireless networking trends. Industry 4.0 use cases that rely on the Internet of Things (IoT), like factory automation, process automation, logistics, and machine-to-machine (M2M) communication, heavily rely on spatially distributed wireless networks (SDWNs), including short-range wireless networks (SRWNs), narrow band-wireless wide area networks (NB-WWANs), 5G-and-beyond networks, and 6G networks [1]. With the help of SDWNs, applications that are geographically distributed and networked may operate more effectively and affordably by enabling seamless wireless communication. These applications have a variety of system requirements, including latency, dependability, battery life, periodicity, range, bandwidth, data rate, and mobility. For instance, while process automation applications are spatially much more widely distributed—possibly over several square kilometers of a chemical or biological process plant, with less time-critical reactions but of greater logical depth—factory automation applications require very fast reactions (down to 10 ms or even 1 ms) and are characterized by a high density of nodes in a production plant. Even though only a single node may be deployed in a system for traditional M2M communication applications like infrastructure monitoring, scalability is extremely high due to the potential to link thousands of devices [2,3]. There are several technological options available to offer dependable wireless communication and satisfy the varied needs of SDWNs.

Functional testing and performance measurements are essential for evaluating and comparing different networking systems. Aperiodic uplink transmissions, burst uplink transmissions, periodic uplink transmissions, aperiodic uplink–downlink transmissions, control loop, aperiodic downlink transmissions, aperiodic downlink–uplink transmissions, and over-the-air firmware updates are just a few examples of the various communication patterns used by applications in the Industrial IoT domain [4,5,6]. Before choosing an appropriate wireless solution, several SDWN technologies must be assessed based on system factors and needs. Performance measurements and functional testing are critical tasks for systematic assessment. Although there are several functional testing and performance assessment approaches, most of them are technology-specific. Additionally, generic test methodologies are based on multiple abstraction layers that model or prototype the system settings, resulting in a lack of uniformity.

In this paper, we present a unified methodology for functional testing and performance measurements, and its prototype implementation specifically designed to address the challenges of diverse System Under Test (SUT) technologies and devices across varying test abstraction levels. Our approach allows for high-level test case descriptions that can be consistently applied across simulation, emulated testbeds, and real field environments. Additionally, we share insights from initial performance evaluations, demonstrating the methodology’s effectiveness in achieving standardized testing and comparability across platforms. This work contributes to a unified and systematic functional testing methodology that promotes consistency, repeatability, and scalability in evaluating future wireless IoT systems.

## 2. Background

### 2.1. Wireless Technologies in Industrial IoT Use Cases

Current 3G and 4G services are unable to provide the wide range of communication demands, including high data rates, high dependability, wide coverage, low latency, etc. However, 5G, with features such as Massive Machine Type Communications (mMTC) and Ultra-Reliable and Low-latency Communications (URLLC), is a viable option for IoT communication requirements in Industry 4.0 [7]. The single-air interface provided by 5G is adaptable, scalable, and capable of meeting the demands of diverse use cases.

To link low-bandwidth and battery-powered devices across long distances, a new class of wireless wide-area networks dubbed NB-WWAN was developed. The Third Generation Partnership Project (3GPP) has defined cellular IoT (cIoT) versions of NB-WWANs, whereas proprietary created technologies like LoRa/ LoRaWAN, SigFox, and MIOTY operate in both unlicensed and licensed bands. Narrow band Internet of Things (NB-IoT) has become the most well-known among them because of its attractive system properties, energy-saving mode of operation with low data rates and bandwidth, and its readiness for 5G use cases.

Our previous studies have evaluated the performance of standardized testing environments for these technologies, as documented in [8,9]. In the current business environment of rapid advancements in wireless technology and a range of wireless solutions, many concerns and challenges occur; these issues and problems can only be answered via thorough testing and measuring. These include identifying the factors that lead to the best performance, determining the requirements for dependable and reliable wireless communication, assessing the suitability of technologies for various applications, contrasting NB-WWAN solutions with cellular solutions like 4G, 5G-and-beyond, and 6G, and looking into how to optimize these solutions to support future applications. It is also essential to understand how standardization efforts might be influenced by testing experiences and how simulation models can be used to forecast a technology’s performance in specific installations. To overcome these challenges, it is vital to use thorough and meticulous testing and measuring methodologies that take into consideration a variety of factors including network design, wireless channel characteristics, interference, and hardware restrictions. By conducting such comprehensive tests and performance measurements, we can gain deeper insights into the functionality of wireless technologies and contribute to their development and successful implementation across a wide range of applications.

### 2.2. Analysis of Related Works of Functional Testing and Performance Measurement Methodologies of NB-WWAN Systems

Many IoT and Industry 4.0 use cases require spatially dispersed NB-WWAN systems. Functional testing is crucial at different phases of the system development process to guarantee the systems’ dependable performance. However, there are several difficulties in functionally evaluating such networks, primarily because wireless communication devices often have limited resources and rely on complex communication layer technologies. The majority of the time, these devices must also cooperate, and since they are frequently installed in unattended settings, stringent dependability standards are introduced. There may be certain firmware-specific mistakes, such as incorrect standards, technologies, or specification interpretation or implementation, as well as generic firmware flaws [10]. Hardware-related issues, including timing errors, resource limitations, and physical malfunctions, also pose significant challenges.

The fact that these wireless networks typically function in complicated topologies, which causes several wireless channel and signal propagation challenges, is another significant problem with them. Additionally, there are other potential causes for wireless communication failure in such systems, including multipath effects, interference, etc. Performance metrics are also necessary for methodically comparing various competing technologies. The growing importance of wireless networks has resulted in the development of numerous testing and verification methodologies, each with unique characteristics and designed for specific test platforms. The most widely used platforms are built on a variety of abstraction levels, including simulation, emulated test environments, and field trials [11]. The typical implementation of these testing platforms makes use of wireless communication channels and networking node prototypes and models.

Around 53% of authors employed simulation in their research, according to a 2016 study [12] of wireless sensor network simulation tools and testbeds. Simulation is, therefore, the research instrument of choice for the bulk of the wireless sensor network community research. Network simulators are often used to compute input data, simulate network behavior, and analyze performance using mathematical models of network functions. There are system-level simulators that aid in the quick assessment of network performance. These simulators employ highly abstracted models that do not take into account hardware, networking, wireless channel, or timing-related behaviours. In-depth simulators address networking behaviors and protocols, but hardware and wireless channel-related aspects are still abstracted.

However, these simulators still abstract hardware and wireless channel-related behaviours. In the past, testbeds have attracted increasing attention, particularly in the case of WSN and IoT research, because the simulation method lacks realism (in testing distributed networking solutions). The examination of various test and performance assessment platforms for wireless systems is the subject of several relevant publications in the literature [10,11]. H. Hellbrück et al. [10] provide an overview of prominent WSN testbeds in Europe and other regions, while other studies [10,12,13] highlight the utility of testbeds in controlled laboratory or field environments using real networking equipment. There are three types of testbeds: the standard laboratory-based testbed, the field testbed, and the testbed as a service (TaaS). Many eminent institutions have testbeds in their labs, which can have substantial deployment costs, maintenance requirements, and environmental dependencies [10,13,14,15]. TaaS offers online access through Web UI and capabilities to design and execute tests by user requirements to get around these restrictions. To meet numerous criteria, including heterogeneity, scalability, portability, federation, flexibility, mobility, interaction, debuggability, software reuse, repetition, and concurrency, several testbeds are available for wireless research [11,14].

The analyses of recent research have explored numerous methodologies to address the complex demands of functional testing and performance measurement across wireless technologies. For instance, the Open-Air Interface (OAI) platform [16] provides a software-based emulation framework, validating the cellular stack under controlled conditions with 3GPP channel models. This platform enables scalable, repeatable testing, though it underscores the need for standardized channel models across platforms to ensure consistent performance metrics. Similarly in other work, Stajkic et al. [17] propose a spatial downscaling methodology that replicates large-scale network conditions in compact testbeds, enabling controlled performance evaluations. However, translating real-world field conditions into lab environments presents challenges, as varied downscaling practices can yield inconsistent performance outcomes, emphasizing the need for a unified methodology. In a related approach, the EuWIn platform [18] and WalT testbed [19] provide controlled environments for IoT and wireless protocol testing, offering reproducible setups but highlighting the difficulty of standardizing tests across different configurations and hardware.

Additional testbed methodologies emphasize the importance of interoperability and unified KPI metrics for systematic evaluation. FIESTA-IoT’s Unified IoT Ontology [20] and the Eclipse IoT-Testware project [21] aimed to facilitate the federation of testbeds and semantic interoperability, introducing standardized data models to support cross-platform comparisons. There are many 5G testbed infrastructures, such as the 5G-MLab [22] and Open5Gaccess [23], which highlight the importance of consistent KPI metrics for reliability and scalability across diverse environments. Recent studies of 5G/wireless testbed implementation work, such as 5GTaaS by Reddy et al. [24] and the TRIANGLE testbed [25], support open-source tools integrated functional testing, while the distributed testbed by Arendt et al. [26] and the SDN-based 5G testbed by Pineda et al. [27] prioritize network safety and flexibility. Notably, Martins et al. [28] focused on energy modeling within virtualized RANs, and Liu and Kalaa [29] introduced the TRUST testbed to evaluate 5G-enabled medical devices in regulatory contexts, both indicating the need for tailored, reliable testing methodologies in mission-critical applications such as industrial IoT. These studies collectively underscore the importance of developing a unified functional testing and performance measurement methodology to ensure systematic evaluation, comparable results, and scalability across diverse IoT and 5G functional test cases.

### 2.3. Need for “Uniformity” in Functional Testing and Performance Measurements

The performance and dependability of wireless communication networks have become a major topic of debate due to the fast growth of wireless technology. A significant issue is the lack of standardization in functional testing and performance measurements, as current solutions are often technology-specific and cannot be applied in a unified context. Despite having a similar underlying architecture, distinct test platforms have varied testing approaches and test case descriptions. This study aims to address the challenge of conducting unified functional testing and performance measurements for NB-WWANs by proposing a unified testbed that enables consistent test case design, execution, and analysis.

In the context of this work, “uniformity refers to the consistency and standardization in the description and execution of tests across various System Under Test (SUT) types and at different levels of test abstraction”. The goal of uniformity is to ensure that, regardless of the specific technology, device, or layer being tested, the methodology for describing and performing the tests remains consistent. This approach simplifies the testing process, making it easier to understand, implement, and replicate across different systems and scenarios. To the best of our knowledge, no relevant publications have addressed the issue of inconsistent testing methodologies and the use of varied test case descriptions.

This issue is important because NB-WWANs sometimes have limited resources and operate in complex network topologies, which complicates testing and dependability requirements. In dispersed systems like the IoT and Industrial IoT, where functions and applications are connected via wireless communication channels, consistent and reliable testing is crucial. A unified testing platform must be able to simulate and test various wireless environments, devices, protocols, and standards, and to gain a thorough understanding of the system’s capabilities and limitations. It also needs to be cost- and time-effective, flexible, better compliant, and standardized.

In this paper, we propose an integrated and unified testbed for NB-WWANs that enables the use of standardized test case descriptions and visualization tools across multiple abstraction levels, including network simulation environments, emulated testbeds, and real field testbeds. By resolving inconsistent testing approaches and enhancing testing efficiency, this study contributes to advancing wireless communication system development and deployment.

## 3. Unified Testing Methodology

This section describes the research objective and methodology, which also includes the extensive research procedure. Specific research questions were developed to align with the study’s aim.

### 3.1. Research Goal

The primary goal of this research is to investigate how functional testing and performance measurement can be made uniform and seamless and to introduce and detail the prototype implementation of a unified functional testing and performance measurement methodology. Additionally, we present early experiences with performance evaluations. The proposed methodology enables the unification of testing approaches and test cases by describing tests at a high level, consistently applicable to various System Under Test (SUT) technologies and devices across different test abstraction levels (simulation, emulated testbeds, and real field environments).

### 3.2. Methodology

In this part, we describe the required terms and methods and introduce a uniform testing approach that enables seamless test flow across various levels of abstraction. This methodology’s primary goal is to make it possible to run unified test cases on many platforms with various levels of abstraction, from simulation to emulated lab testbed to field testing. For the intended system/technology under test, the tester must be able to conduct seamless functional testing and performance assessment across a variety of platforms, including network simulators, emulated testbeds, and real-world field test environments. We propose a spiral model-based testbed approach that integrates testing across all stages of the System Testing Life Cycle (STLC). The central concept of this approach is to utilize uniform test cases when evaluating the system on different test platforms—network simulation, emulated testbeds, and field testbeds—throughout the STLC. The research methodology employed in this study was derived from the hypothetico-deductive method [30]. The research methodology is carried out as a scientific inquiry, starting with the development of a fictitious research objective, and progressing to the goal’s augmentation by testing and discussion of each research contribution. In observing the data gathered during the experimentation process, there was no bias present. The solutions were benchmarked and published to emphasize the research contributions with reliable confidence in the observed findings. Figure 1 depicts the research method and a summary of the unified approach for conducting functional tests across various test platforms by managing every piece of the test environment using the same test execution tools.

The following are the main steps of this unified multi-abstraction layer testing methodology.


**Step 1: Unified Test Description**


The first step involves defining the use case, test specifications, and unified test strategy. To ensure consistent functional testing of spatially distributed wireless systems across multiple platforms, a comprehensive test strategy is required. This strategy should encompass clear objectives, scope, methodologies, and detailed test cases and scenarios described in the test plan. A consistent test data and control format that can be utilized across all types of System Under Tests (SUTs) and Test Platforms should be used in the test cases to make them easy to run on various test platforms. Tests for a single distinct idea (such as transmitting UDP packets on an NB-IoT network) must be parameterizable as much as possible within the concept’s scope. At the level of the exchanged abstract data messages and signals, clear definitions of interfaces between the test system and the system under test are established.


**Step 2: Unified multi-abstraction-layer testbed setup**


The second step is to build up the unified testbed, which combines several abstraction-level-based testing platforms, including field testbeds, emulated testbeds, and network simulation. It is important to create unified test interfaces for the various test platforms and systems under test (SUT). The test focuses on the SUT, which receives input test messages (often referred to as “stimuli”) designed to prompt specific behavior. The test description will change since different devices have various interfaces. As a result, the test system has to be integrated by using integrated test interfaces or command sets for various devices. Additionally, the test system should not be dependent on a particular hardware interface. It should support a variety of hardware interfaces (e.g., serial and TCP/IP), and switching between these interfaces (e.g., from serial to TCP/IP) should be straightforward.


**Step 3: Uniform Test execution procedure and use of common toolsets**


The third step involves executing the test cases on each test platform using the same test automation framework, tools, procedures, and data analysis techniques. This ensures the comparability and significance of test results across test platforms. The tests should be specific enough to test individual components comprehensively, while the test environment should define the tests as consistently as possible across platforms.


**Step 4: Uniform Test visualization/analysis**


Gathering and analyzing test data from each test platform is the fourth step. The results must be visualized in a uniform manner to ensure findings are precise and easy to compare. Using standardized data formats and protocols, as well as automated data gathering and analysis tools, are some examples of this.


**Step 5: Spiral model-based iterations and Optimizations**


To make sure that the test objectives are satisfied and that the findings are consistent, step 5 of the suggested testing methodology entails constant monitoring and review of test results across all testing techniques. To get ready for the next round of testing, the assessment phase informs the improvement and optimization of test cases, as well as the testing environment and tools. Through iterative testing and performance measurement improvement, this spiral model-based technique ensures systematic testing.

For the first level of tests and analysis against system requirements and design, simulation model-based test platforms are employed in the testing process. The majority of system components, such as the hardware of MCUs and transceivers and wireless propagation channels, are abstracted by the simulation platform. Following the first assessment, testing is carried out in an emulated testbed where the wireless channel characteristics are abstracted, enabling wireless channel abstraction and test repeatability. Once the operational prototype is available, testing is carried out as field testing while taking into account all actual wireless propagation effects.

The suggested technique takes into account the implications of various abstraction levels and chooses appropriate test platforms for various test kinds across the testing life cycle while reusing identical test cases. This method provides a thorough understanding of the performance and behaviour of the wireless system that is being tested in various circumstances, enabling the detection and correction of problems prior to deployment. The testbed must specify a method of characterizing and parametrizing networks on various tiers in order to govern all parts of the test environment from a single test execution tool. In order to provide more flexible use cases, additional devices must also be programmable from inside the test case. These additional devices can include a signal generator for noise production, a custom test system emulation, or various measurement and analysis tools.

### 3.3. Unified Test Description Methodology

Testing and Test Control Notation version 3 (TTCN-3) is a standardized testing language designed for communication systems testing [16]. It is suitable for various testing requirements and supports both static and dynamic setups. TTCN-3 is designed for readability, allowing non-programmers to create, read, and comprehend test cases. This transparency promotes collaboration among stakeholders, including testers, designers, and system specialists. Its essential principles include components, ports, and messages, which are concurrent test system entities that perform test operations [16]. Titan is a toolset that provides an integrated environment for TTCN tests, facilitating the development and execution of test suites. TTCN-3’s unique traits and advantages make it the preferred choice for the unified test description methodology.

The uniform Test Description methodology seeks to provide a uniform approach to describe tests across various System Under Test (SUT) types and degrees of test abstraction. The technique is broken down into three levels, as shown in Figure 2:

Layer 1: Oversees the overall testing process.Layer 2: Implements abstracted behaviour through related test interfaces and functions.Layer 3: Provides specific device behaviour implementations.

**Layer 1: Unified Test Description and Management:** This layer defines, controls, and executes all tests consistently, ensuring uniform management of testing tasks. It uses an abstract test case structure with phases for the preamble, test purpose, and post-amble using identical configuration files to greatly parameterize the setup parameters. The test control section coordinates the entire test and consists of cohesive test execution modules that describe a test at a high level, including which components to use, how to connect them, when to start them, how the network looks and changes, which nodes to use, and how to coordinate them in the system. The Titan configuration file, which corresponds to one module and parameterizes it for several profiles, may be used to configure the test system. The objective of the abstract test case is to test just one item and to comprehend it clearly while describing the test as similarly as possible across multiple test platforms.

**Layer 2: Implementing Abstracted Behaviours:** Through related test interfaces and the execution of functionality, Layer 2 implements abstracted behaviour. Ports that only accept particular message types make up the stated components in the TTCN-3 schema. The fundamental operations of the majority of components may be abstracted using these message types. In order to standardize the functionality of a technology, interfaces are also established for the SUT control portion. All technologies and abstraction layers can use the same abstract behaviour description. The implementation of a uniform command set simplifies testing operations and enhances repeatability across diverse SUTs, ultimately leading to more efficient and reliable testing methods.

**Layer 3: Specific Device Behaviour Implementations:** Through classes that inherited the functions described in Layer 2, Layer 3 offers individual device behaviour implementations. The functional modules are made up of a number of functions that call the concrete implementation using the internal device object. Depending on the device being utilized, different device controls are implemented. The equipment normally responds to particular requests within a predetermined amount of time. The controller verifies the outcomes, ensuring the test proceeds only when the expected results occur within the allotted time. The test system must also look for asynchronous communications and error messages.

In general, the unified test description methodology offers a highly customizable and consistent way to describe tests across various SUT types and at various test abstraction levels. By utilizing abstracted behavior descriptions and standardized command sets, the methodology enhances the efficiency and reliability of testing procedures.

## 4. Prototype Implementation

To develop, test and evaluate the unified multi-abstraction-level testing methodology, we mostly adhered to the prototyping technique. The prototype implementation details of the unified testing methodology proposed in the previous chapter are described in the section. The unified testing system for unified test case description and execution is built using the Eclipse Titan framework version 8.2.0, with certain restrictions imposed by Titan and TTCN. The test system architecture consists of the TTCN unified test case description concept with parametrized test cases, components for running test cases, and functions for describing underlying detailed technical functionality. The test system is then integrated into three prototype test abstraction levels such as network simulator (NS-3), emulated testbed (using in-house developed Automated Physical TestBed [14]) and field test environment (by integrating various NB-WWAN devices under tests and measurement devices).

### 4.1. Unified Test System Architecture

Figure 3 depicts the architecture of the unified test system and the key concepts that were utilized to include several abstraction levels in the Titan framework, including network simulation (NS-3), the Automated Physical Testbed (APTB), and the field-testing environment. The integrated and unified test system consists of the run-time setup, test suite, and a server to store and process results. It also has a unified test system interface.

As provided above, the main elements of the integrated and unified test system primarily include:

#### 4.1.1. Unified Test System Interface

The goal of the unified test interface is to provide a user-friendly and generic interface for configuring and running test cases. A configurator component enables the tester to view configuration options for each technology and abstraction layer and appropriately sets up the Titan configuration file. The Executor component is in charge of displaying to the tester the test cases that are accessible, writing the test cases to run into the configuration file, and running the tests.

#### 4.1.2. Unified Test System Components

The Unified Test System consists of three primary components: the Run-Time Configuration, the Test Suite, and a Server for storing and processing the results.

**Run-Time Configuration:** determines what device will be tested, what configuration will be used, what test case will be tested, and at what abstraction layer.**Test Suite:** comprised of Runners and the Main Component (MC). Test case flow is described by the MC, who also coordinates the other elements. Runners are components used in unified test execution. They include.○**SimRunner:** The SimRunner component is used to communicate with the simulation server’s REST service. The SimRunner configures a test case, launches it in the simulation server, and then retrieves the results. Through the creation of a test verdict and local storing of the results, the simulation runner provides feedback to the tester. The Simulation REST-API defines the interface with the simulation server, offering endpoints for starting parameterized simulations, creating projects and subprojects, and retrieving results.○**APTBRunner:** To set up the channel conditions, the APTBRunner communicates with the APTB. The APTB runner manages the translation process from channels to various RF elements and connects with the APTB via the APTB’s REST-API. This provides a simplified interface for configuring and reading channel properties, enabling control of RF elements assigned to element controllers and groups via API endpoints.○**NodeRunner:** NodeRunners are used to interface uniformly with various devices or implementations being tested. As a result, each device in a particular device class implements the behaviour given as a class interface. A configuration option is used to enable the usage of several devices, making it easy to select the device without having to recompile the system.


### 4.2. Integration of Test Abstraction Levels to Unified Test System

The network simulator (NS-3), the internally developed emulated testbed (Automated Physical TestBed), and the field test environment (with various NB-WWAN technologies devices under tests and measurement devices) are the three different test abstraction levels covered in detail in this subsection. We provide clear guidance on the integrative stages and processes necessary to successfully merge these parts into the unified test system by breaking down this process step-by-step like a recipe.

#### 4.2.1. Network Simulator (NS-3) Integration

Due to its adaptability, open-source status, and pre-existing models for crucial wireless technologies like NB-IoT, Network Simulator (NS-3) has been chosen as the appropriate simulation platform to integrate with the unified test system. A discrete-event simulator for Internet networks called NS-3 has undergone numerous updates and releases, each of which included a new set of features and functionalities [15]. The following are the primary criteria taken into account for NS-3 integration into the unified test system:(1)**Reproducing unified abstract test cases** on the NS-3 platform to simulate diverse scenarios using available models.(2)**Initiating simulations** through an execution script from the Unified Test System.(3)**Building and creating simulations** for a variety of test cases using preset simulation models available in NS-3.(4)**Simulating the interaction of multiple devices** based on the configuration options provided by the test system.(5)**Retrieving data from the simulator** for unified analysis and display.

The key component of the system topology shown in Figure 4 is a simulation backend. This takes domain-specific language simulations but does not run the simulations; instead, it creates an SSH connection with the simulation runner. The NS-3 simulation environment is hosted by the simulation runner. When the simulations are complete, the simulation backend may read out the results using SSH, analyze them, and save them in a database. The source is irrelevant as long as the simulation description follows the domain-specific description. Potential sources include but are not limited to, a simulation frontend or a description written in the TTCN-3 description language from the unified test system. In addition to simulation, the simulation environment also includes the creation of projects and subprojects, which are used to plan test campaigns and parametrize simulations. Access to the simulation environments’ functionality is made possible through a REST API. This API is used to create new projects and simulations, carry out such simulations, and obtain the results of those simulations. **The steps for testing with these integrated NS-3 simulations include**:

**Creation of NS-3 simulations in the simulation backend:** Different capabilities and use case scenarios are created to simulate various NB-WWAN technologies. Aspects including network configuration with User Equipment (UE) and Base Stations, radio propagation circumstances, device placement, communication patterns, and packet transmission monitoring are all taken into account in the simulations. Additionally, they make it possible to measure various network events, assess those measurements, and save the results.**Execution of Simulations by the Simulation Runner:** This is used by both the TTCN3-Testsystem and a frontend that enables direct user participation. The Test system uses Titan’s HTTP test port to implement the various REST Endpoints. When using the REST-API, projects follow the path depicted in the sequence diagram in Figure 5. New projects and simulations are created, carried out, and their outcomes are accessed via a REST-API using the TTCN-3 test system. This interaction takes place in the following order: setting up projects and subprojects, sending parameter combinations, starting a simulation run, checking on the progress of the run, and recording the outcomes locally and on the simulation server.**Results Visualization and Analysis:** The simulation server is equipped with an automatic visualization and analysis tool that gives the user immediate access to the outcomes of a single test run. Overall, through numerous simulations in a single test system, this integration encourages a methodical and comparative analysis of various NB-WWAN systems.

#### 4.2.2. Automated Physical TestBed (APTB) Integration

Before detailed implementation and during the development cycle, these communication technologies must be tested and validated. For this, our unified testing methodology utilizes the in-house developed Automated Physical TestBed (APTB) and integrated with the unified test system, as illustrated in Figure 6, in conjunction with emulation platforms and other necessary tools for analysis. These emulate RF environments that are close to real-world conditions, enabling controlled and systematic testing. Further details on the APTB design and implementation are provided in our previous works [8,9]. Integrating APTB with a unified TTCN-3 test system, along with dedicated automated control interfaces, allows seamless configuration of diverse spatially distributed topological setups and channel conditions. The APTB architecture prioritizes modularity, extensibility, scalability, portability, and ease of setup and dynamic configuration, fostering adaptable testing across various conditions.

##### Software Architecture of APTB Integration to Unified Test System

The overall software architecture of APTB, depicted in Figure 6, consists of a Main Controller, multiple Element Controllers, and communication management via Node–RED. In the implementation, a Raspberry Pi serves as the Main Controller, running multiple Docker containers to streamline control and integration. The Element Controllers register RF element configurations with the Main Controller, which manages these configurations through a centralized Node–RED interface. This platform arranges data flows between TTCN-3, the APTB, and external scripts, with all metadata stored in a MongoDB database managed by Mongo Express. Additionally, through Portainer, a container management solution that provides the viewing of the containers, images, volumes, and networks, all of the Main Controller Docker containers may be handled.

##### APTB Channel Configuration and Characteristics

The APTB supports bidirectional channel configurations, consisting of multiple nodes (DUTs) and channels that connect the nodes. APTB channels are designed to ensure symmetry in both communication directions, effectively modeling an undirected graph where edge values are set by configurable parameters. Channel characteristics, illustrated in Figure 7, include options for attenuation, on/off switching, multipath settings, delay lines, and noise control. Each of these parameters is implemented through distinct HF elements, each controlled by its own Element Controller.

##### APTB Component Interactions and Control

The test system’s AptbRunner components continuously monitor for messages from test cases, conveying abstract descriptions of APTB configurations. These messages undergo processing within the ChannelAPI and AptbAPI modules before dispatch through HTTP requests to the APTB, as shown in Figure 8. The AptbRunner accepts a variety of configuration formats, such as single-channel setups, node-to-node configurations, and time-based configurations.

An example APTB configuration is structured as Listing 1:

**Listing 1:** APTB configuration formatAptb:= { hf_elements:= {  {channel_id:= “13”, switch_state:= true, attenuation_db:= 5, delay:= omit, multipath:= omit},  {channel_id:= “511”, switch_state:= omit, attenuation_db:= 5, delay:= omit, multipath:= omit},  {channel_id:= “26”, switch_state:= false, attenuation_db:= 5, delay:= omit, multipath:= omit} }}

##### APTB Unified Control and API Abstraction

The control software of the APTB is implemented in a layered API structure to manage communication and operations across its diverse RF components. At a topmost level, the ChannelAPI module provides an abstract interface for REST API requests, providing a systematic mechanism for setting up and manipulating channels between nodes. This abstraction allows the user to configure channel parameters, such as attenuation or delay, without needing direct interaction with each hardware component and makes the test environment easily reproducible.

The AptbAPI module, on the other hand, operates at a more granular level, managing the specific REST commands for each type of RF element within the system. This design approach encapsulates functionality within distinct API calls, ensuring that each RF element (e.g., attenuators, delay lines) is individually controlled according to its parameters and test case operational requirements. For example, the system sends a targeted request that modifies only the relevant RF element’s parameters, while maintaining the overall channel configuration, to adjust the attenuation value on a specific channel. This modularity allows for systematic control over the APTB, which facilitates real-time reconfiguration and nuanced adjustments during testing phases.

##### APTB Unified Configuration and Mapping

Configuring the APTB channels necessitates a careful manual setup process, primarily due to the structural asymmetry in the arrangement of RF elements. This asymmetry means that connections between nodes may require distinct RF elements to achieve the desired channel characteristics, making a systematic channel characteristic essential. The mapping process involves assigning each channel to specific RF elements, which are then controlled to achieve the defined channel attributes (e.g., attenuation, on/off states, multipath, and delay options).

Once the channels are parametrized, the test system must interpret this parametrization and translate it into REST requests, effectively linking the logical channel configuration with the physical RF elements. A precise, node and channel-based notation system is implemented on the GUI side and is used to represent each channel, enabling straightforward identification and differentiation between channels based on their node associations within the APTB structure. This notation enhances usability, as it allows the testbed users to manage complex configurations by referring to easily identifiable node names and attributes rather than raw element IDs. This systematic testing approach ensures that each channel is distinctly mapped to its RF elements, thus facilitating accurate, repeatable testing across a range of communication scenarios.

#### 4.2.3. Integration of Real-World Devices and Measurement Equipment for Field Tests Using Unified Test System

For real-world field testbed implementation, we gathered and integrated the commercially available NB-WWAN, 4G, and 5G network communication devices into the Unified Test System as illustrated in Figure 9 to analyze the effectiveness of various competing wireless networks. These devices can be separated into two categories: base stations, which are frequently referred to as gateways, and various end devices.

##### Unified Test Interface for DUTs and Control by Unified Test System

The goal of unified test interfaces is to enable the execution of test cases in a technology-independent manner, ensuring consistency across different devices and technologies (Figure 9 summarizes the devices used for these prototype evaluations). The test cases are designed to be as generic as possible, allowing them to be executed independently of the specific technology. However, for certain technology-specific tests, additional details and functionalities can be accommodated through corresponding interfaces. To interface with different devices that have different command sets, a **unified test interface with a mapper component** is utilized. This component allows the use of generic commands that can be adapted to the specific command sets of each device. By providing specific implementations for each device, the unified test system can interact with the DUTs effectively.

**Hardware and Application-Level Interfaces:** At the hardware interface level, the unified test system supports various communication protocols, such as serial and TCP-based interfaces. The selection of the appropriate interface is determined by the configuration file. Each device implementation utilizes methods for sending and receiving data via these interfaces. The test port records communication timestamps, capturing the timing of message transmission and reception. At the application level, the received messages are checked for successful transmission and the type of message, enabling the system to generate appropriate events for further analysis.

For example, the unified test interfaces in NB-IoT provide a standardized approach to interact with different devices under test, ensuring seamless integration and consistent test methodologies. By unifying the interfaces, test cases can be developed and executed independently of the specific technology, promoting interoperability and efficient testing processes. This approach streamlines the testing of NB-IoT systems, simplifies test case development, and allows for comprehensive analysis and evaluation of performance and functionality.

##### Integration of Measuring Devices and Analysis Tools to Unified Test System

In addition to the automation of Devices Under Test (DUTs), the integration of measuring devices and analysis tools is crucial for comprehensive testing and analysis. These devices provide the capability to measure and analyze various network parameters, signal quality, coverage, latency, and other relevant metrics. To ensure seamless integration and automation, standardized interfaces and protocols are utilized, and a detailed overview of the flow of unified interfaces and controlling architecture for measurement devices and DUTs. Measurement devices often employ the Standard Commands for Programmable Instruments (SCPIs) protocol as the digital interface. As part of the automation efforts, the SCPI interface of the proprietary tools like Rohde and Schwarz CMW500 communication tester is implemented in TTCN-3. A SCPI controller is defined to handle the SCPI protocol, and a class for the CMW500 is created to implement its functionality. While not all functions of the CMW500 are implemented due to their abundance, essential capabilities such as setting network parameters, activating cells, and introducing noise on the downlink channel are supported. The CMW500 component can be controlled from test cases through a defined port interface, allowing for the configuration of technology settings and noise configurations.

To enable analysis and measurements in the test system, various measurement devices and analysis tools are integrated. These devices range from generic measurement devices such as signal generators, and signal and power analysers to specific measurement tools tailored for NB-IoT testing. The integration of these devices into the existing infrastructure provides a fully automated test and measurement environment. For example, in our recent NB-IoT test campaign, we utilized various testing tools such as the CMW500 with specific testing options [26], Anritsu MT8821C with dedicated testing options [27], and Keysight’s Nemo analysis Tool [28]. These measurement devices and analysis tools come with their own specific interfaces, which are seamlessly integrated into our unified test system. Through the standardized interfaces, measurements related to signal quality, network coverage, latency, and other relevant metrics can be configured and retrieved. This integration enables comprehensive measurement and analysis of NB-IoT network parameters, ensuring accurate evaluation of performance and functionality.

### 4.3. Unified Test Suite Implementation

An in-depth discussion of the Unified Test Suite Implementation’s layers and individual components is provided in this Section. The unified description of tests is managed by Layer 1, also referred to as the Unified Test Description. It establishes an abstract test case with a preamble, a test purpose, and a post-amble phase and heavily relies on configuration settings parameterized through identical configuration files. The three main components of this layer are Test Control, Test Configuration, and Abstract Test Case. This chapter also provides a practical scenario using the “Periodic Uplink” test case as an illustration. The system implements an abstraction of behaviour in Layer 2, or Abstract Behaviour, by using corresponding test interfaces and the execution of functionalities using components and interfaces.

#### 4.3.1. Layer 1—Unified Test Description and Management

By describing and controlling tests uniformly, Layer 1 controls the unified testing task. The configuration settings are strongly parameterized using identical configuration files, and a preamble, test purpose, and post-amble phased abstract test case is created. This layer primarily consists of the Test Control part, Test Configuration part, and Abstract Test Case part as shown in Figure 10 and detailed as follows:

**Unified Test Control Part:** The test control part orchestrates the overall test and consists of coherent test execution modules that describe a test at a high level, including which components to use, how they should be connected, when they should be started, how the network should look and behave, which nodes should be used, and how they should be coordinated. Additionally, it describes the intended test behaviour.

**Test System Configuration:** The Titan configuration file, which corresponds to one module and parameterizes it for various profiles, may be used to configure the test system. Device-specific parameters, wireless parameters, and technology-specific parameters can all be set in the existing framework. Without specifically altering the configuration file, the test may be configured using profiles. The pertinent parameters, such as the test application, devices utilized, technological setup, network configuration, and wireless environment, differ depending on the layer being evaluated. For instance, concrete devices must be mentioned in the APTB and field layer, while other parameters are used in the simulation to describe these devices.

When it comes to technology, all technologies employ the same characteristics. Nodes are put at precise locations in the field and simulation, but in the APTB, only the channel conditions change. In the simulation, path loss models and building models are utilized to represent the wireless environment, and the wireless environment parameter is used to summarize the node position and the wireless environment. The test system configures the parameters, which are generated by an external component, using Titan’s configuration file. The test suite’s unified configuration types include parameterizing test ports (such as node IP addresses), selecting the SUT type, network, and test case, deciding where to save the data-acquisition file, and configuring test and technology parameters (such as packet counts, operation modes, and test cases).

**Abstract Test Case Part:** The goal of the abstract test case is to express the test as consistently as possible across different test platforms while being as explicit as is required to test only one item and comprehend it. Additionally, using the accompanying configuration file, it ought to be extremely parameterizable. The test case goes through three phases: the preamble, the test purpose (primary test), and the post-amble to converge the test description. A test case must always begin with a preliminary part that correctly configures the SUT and the testing environment.**Example Test Scenario: “Periodic Uplink” Test Case:** To detail this concept of unified test description, let us consider an example test scenario of the “Periodic Uplink” test case and demonstrate how it can be described to run on multiple DUTs and test abstraction levels. The “Periodic Uplink” test case focuses on assessing the system’s ability to establish network registration and maintain an uplink connection between the endpoint devices (EDs) and the base station (BS) in a periodic manner. In the context of using an NB-IoT device under test for a periodic uplink test case, the test Control Part would outline and manage various aspects of the test. Figure 11 details the concepts of this example abstract test case with a sequence diagram of the entire test procedures, corresponding implementation details on the three stages for a unified abstract test case and outlining how it is mapped for uniform execution to different test abstraction levels.

#### 4.3.2. Layer 2—Abstract Behaviour

Layer 2 of the proposed system implements abstracted behaviour via corresponding test interfaces and the implementation of functionalities. As shown in Figure 12, several components and interfaces are developed to abstract common functionality. Each component represents a component type with specific capabilities, playing a unique role in the system:

**MC (Main Controller):** This is the principal component controlling various abstraction levels. It is responsible for coordinating all the other elements and directing the overall testing process.**Writer:** This component is responsible for the logging of concurrent activities across various components. It receives specific Write-Events and calls C++ functions to store the data safely, thus providing a comprehensive record of testing activities and outcomes.**Common node:** As the name suggests, this component embodies common concepts and ports, including a node control port (for control from the Main Controller) and a writer port (to write asynchronously to the writer component). It acts as the baseline model for all nodes, defining the minimum set of interfaces required.**Node:** This component is specifically designed to interact with real devices. It has TCP and serial interfaces for diverse communications. Furthermore, it maintains an internal state regarding the device in use and the protocol employed, enabling it to select the appropriate functionality as needed.**UDP-Server:** This node type has an added UDP port, allowing it to communicate using the UDP protocol, thereby increasing the communication possibilities within the testing environment.**HttpNode:** This component has an HTTP port, enabling it to communicate with external systems via HTTP protocol, thus enhancing the system’s interoperability.**AptbNode:** This node type possesses an additional APTB Control message, as described earlier, to manage the APTB. This gives it the capability to control and interact with APTB-related tests.

**Standardized Interfaces and Implementation:** It is worth noting that these interfaces merely define the blueprints for device capabilities. They are then implemented in the actual devices, using the TTCN interfaces to ensure that each device must fully implement its associated blueprint. This standardizes the implementation and ensures uniformity across multiple devices, contributing to the overall effectiveness and dependability of the test processes.

Ports that only accept particular message types make up the stated components in the TTCN-3 schema. The fundamental operations of the majority of components may be abstracted using these message types. To standardize the functioning of a technology, interfaces are also established for the control portion of the System Under Test (SUT). All technologies and abstraction layers can use the same abstract behaviour description.

Examples of abstracted modules are as follows:Node: The abstraction of node activity without regard to a particular technology.Test ports: Test ports are used to interface with SUTs that are stored in the Titan-Gitlab repository.Test layer: abstracted behaviour of the test layers, such as field testing, simulations, and Automated Physical Testbed (APTB).Utilizing external C/C++ code, such as timestamping, CSV writer, and *.asn: Generic Command and Event Specification, are examples of common utility and external modules.

##### Unified Test Interfaces for SUTs

To provide a consistent approach during testing, it is required to abstract the commands and interfaces utilized by the System Under Test (SUT). Using a command set that abstracts the functions of several SUTs can help with testing and boost productivity in this respect. As an illustration, the functionality of the SUT is abstracted at the device level in the case of NB-IoT (refer to Figure 13). A command that corresponds to the specific NB-IoT modem utilized is issued when a reference to the NB-IoT interface is called during the execution of a particular function. Despite the particular device being tested, this enables the usage of a common command set.

The setup function needs to be called before running the instructions to utilize the right test port and device. The interface type corresponds to the appropriate device implementation class, and the configuration file defines the device and test port interface to be utilized. Overall, the implementation of a uniform command set can streamline testing operations and enhance repeatability across many SUTs, eventually resulting in testing methods that are more effective and dependable.

##### Unified Abstracted TTCN-3 Component Types for Technology Functionalities

The Test System is made up of many TTCN-3 Test Components, each of which is in charge of one job that may be carried out in parallel and is connected to other components via test ports. For the port definitions to connect with the components, message types must be established. The Titan framework naturally supports parallel test execution and event queues. Figure 14 depicts the components and their ports. Execution of test cases on Layer 1 and other components is managed by the MC component. It has ports for controlling Aptb, Simulation, and Node Components that carry out device-testing behaviour.

The Test Components can communicate events, their decision, and a reference to a Common_Node_CT back to the MC using the NodeCnt_PT port, as shown in Figure 14. The Test Components can also initiate and halt test execution via the NodeCmd. Depending on the needs of the user, the APTB port specifies a variety of message formats to control the APTB channels. For the port types towards external components (udp, tcp, http, serial), the test ports that are already accessible from the Titan gitlab repository are utilized. To abstract the behaviour of multiple capabilities, TTCN-3 Test Components are built. Here are a few instances:

A node in a test might be an SUT or an endpoint, such as an udp/tcp server or an MQTT broker. This component abstracts common node behaviour.

To receive start/stop instructions, and general directives, or to relay back the outcome of a behaviour step, it communicates with the MTC via the NodeCnt_PT.Several Node-Types, including TCP, Http, Node, and UdpServer, derive from this component. The SUT’s controlling node is referred to as an abstracted specific node component.It has a TCP port and a serial port for communicating with the SUT, and the configuration file must specify the specific port to be utilized.From this type, technology-specific components must be derived.Abstracted UDP Server Component: Inherits from Common_Node_CT and extends it by udp port to establish a UDP endpoint for the tests.Abstracted TCP Node Component: Inherits from Common_Node_CT and extends it by tcp_port, used by all components that need a TCP port, such as SCPI measurement devices.Main Component: Component type for Layer 1 that communicates with all components.

#### 4.3.3. Layer–3—Concrete Functionality Implementations

Classes that inherit the features described in Layer 2 are used to create particular device behaviour in Layer 3, the flow relation between the abstracted and concrete implementation is described in Figure 15. Due to the necessity of running numerous components simultaneously, single functions are used to start tiny “atomic” behaviours. Every technology-specific behaviour is put into practice in a different module. The functional modules are made up of several functions that call the concrete implementation using the internal device object. Depending on the device being utilized, different device controls are implemented. The equipment normally responds to particular requests within a predetermined amount of time. The controller is in charge of confirming the outcome and making sure that the test only proceeds when the anticipated outcome occurs within the allotted time. The test system must also look for asynchronous communications and error messages. For instance, the functionality module provides the function to set up a UDP socket using the device object, which entails generating and connecting the socket, when building a UDP socket. The general send command, which is in charge of using the selected interface and timestamping, is used in the concrete implementation to transmit a particular command. A generic receiving function is used to verify the anticipated event, and the test only continues when the socket has been successfully constructed.

Asynchronous responses can also happen in addition to instantaneous answers, independent of whether the controller issues a command. Users can explicitly activate call-backs to catch such events. Call-backs can only be defined in the test system because the emphasis is on black box testing of different devices. Some events can happen at any time, in contrast to device events, which are instantaneous replies, and they are detected by the test system using a function that specifies the event type and potential action. The act_cbk command may be used to activate preset call-backs in an IUT so that it can be tested. For instance, depending on the base station setup in NB-IoT, a function may be notified of a modified PSM status, and it is the responsibility of the function to recognize the change and update its internal state.

### 4.4. Unified Test Case Execution and Analysis Across Multiple Abstraction Levels

The processes necessary for the coordinated execution of test cases across various abstraction levels are described in Figure 16. First, test components are developed and configured to communicate with various test system components. For example, the NodeRunner component is configured for a specific device to ensure the correct instructions are issued. The right starting state is then configured by delivering the required commands to the external test system components, such as the Simulation server, APTB, or SUTs. For instance, customizing the APTB could entail adjusting the channel attenuation, but configuring SUTs might involve turning on a certain frequency band. In contrast, the Simulation’s settings are established in the step just before the test is launched.

The primary test starts as a series of function calls with each function explaining a specific feature of the capability of the technology after all the components have been set up. The test system handles events that arise during test execution, and pertinent data are logged. When a simulation is engaged, the simulation server automatically delivers the results back to the SimRunner Component and puts them in the database. Setting the decision and preserving the data are the responsibilities of the SimRunner. The default state of the device is restored once test cases have been run, components are shut down, and all data and logs are saved. A Mapper Component is used to communicate with multiple devices that have distinct command sets.

With the help of the Mapper Component, it is possible to interface with many DUTs using generic commands, but this requires both a general command definition and a device-specific implementation. The Mapper is now incorporated into the NodeRunner components. The main component (MC) coordinates the test system’s overall behaviour. Titan employs Parallel Test Components (PTCs) to distribute, log, and interact with these components to operate many devices. One NodeRunner per device, parameterized for a particular kind (such as ublox-sara-r5), is used for separate control of each device. The APTBRunner is used to operate the APTB, which is used to regulate channels between nodes.

An APTB-API is designed as an abstraction to these RF elements since the APTB is equipped with certain parameterizable RF elements. It is required to comprehend the test system’s usage, including the configuration file format, how to activate test cases, and how to launch the program using the command line. An essential part of automating test system execution is the front end of the tests. To increase usability, a REST API was created that outlines the main features of the test system. These features include selecting the abstraction level (field, emulation, and simulation), a technology and device, one or more test cases, predefined profiles, configuring data traffic and environmental conditions, automating test execution, and providing feedback on the results of completed tests. Figure 17 shows the front end that was built. The interface presents all choices as opposed to having to manually update the configuration, preventing the selection of test modules and incompatible technologies.

## 5. Performance Evaluation

This section contains a case study evaluating the effectiveness of our suggested multi-abstraction-level, unified testing methodology. To determine the methodology’s consistency and efficacy across various abstraction levels, it was tested in three separate environments: simulation, emulation, and field testing. These were chosen as the test campaign environments. The case study was initially intended to record a thorough baseline of network performance measurement under diverse real-world circumstances. To comprehend how various factors might affect network performance, we described the distinctive characteristics of each testing set. We developed a mapping technique to enable seamless transitions by running the same test cases across different test abstraction levels. We then continuously monitored the state of each environment to obtain the most recent performance data. The findings of this comprehensive research serve as the basis for further, more in-depth testing.

Three separate contexts were used to carry out the testing campaign: a field test in the real field environment at the University campus, an emulation environment using the APTB channel characteristics, and a simulation environment utilizing NS-3 tools and models (refer to Figure 18). Industry 4.0 use case ecosystem is used to characterize the test cases uniformly. Each environment underwent testing using a typical Industry 4.0 communication use case, which involved successfully attaching a user device (UE) to a base station (BS) and then periodically sending uplink UDP packets. Using the same test case across different levels of abstraction enabled a time-effectiveness and flexibility analysis of our unified testing methodology. This analysis provided important insights into the subtle distinctions between the various levels of abstraction.

The findings highlighted the consequences of abstraction in Figure 19 by revealing significant variations in the key performance indicators (KPIs) throughout the simulation, emulation, and field tests. However, the results from the simulation environment nearly matched those from the field tests conducted in the actual world, proving the effectiveness of the system for conducting carefully monitored, repeatable trials. To further assess the success of the methodology, a thorough comparison of the Reference Signal Received Power (RSRP) and Signal Noise Ratio (SNR) between field and emulation conditions was made. As shown in Figure 20, the SNR in the emulation environment was found to be consistently greater than in the field tests, indicating that there were no real-world elements present in our emulation system, such as multipath propagation or background noise. However, by modifying the RSRP in the emulation environment, we were able to obtain nearly equal median RSRP values in both environments.

**Key Insights from the Case Study:** The case study results show the unified testing methodology’s impressive efficacy at various levels of abstraction. The testing circumstances varied, but the process was applied consistently and produced trustworthy findings. The methodology’s capacity to smoothly switch between several levels of abstraction and its ability to be successfully deployed in a variety of test contexts without exposing the integrity of the testing process or the validity of the results were both demonstrated. However, a few prerequisites were found to guarantee the methodology’s applicability at various levels of abstraction. These include undertaking thorough result analysis, assuring correct abstraction and dependable emulation, providing consistent parameter settings throughout testing environments, and including actual field-testing situations. The effectiveness of the methodology could be increased by putting more emphasis on precisely simulating actual situations, such as reflections, multipath components, and interference sources. Understanding the applicability and efficacy of various test cases across multiple abstraction levels for the NB-WWAN depends on test coverage analysis and abstraction layer mapping.

With the aid of this knowledge, the ideal abstraction level for each functional test case and performance metric can be determined, enhancing the testing strategy as a whole. For instance, network simulations may not fully reflect real-world conditions due to simplified models, although they are good for first functional tests. The majority of test cases can be used using network emulation since it balances controlled settings and real-world characteristics and typically produces great results. Field tests, though they might be difficult and time-consuming due to unpredictable real-world circumstances, offer the most realistic testing conditions. The limitations of each platform are often a result of these fundamental qualities; for example, outdoor experiments may include uncontrollable variables that affect results, while emulations may not be completely accurate in simulating specific situations. As a result, this mapping aids in determining the best degree of abstraction for each test case, guiding a thorough testing strategy that guarantees complete coverage for the NB-WWAN network. The case study findings reported in this paper serve as a first assessment of our unified multi-abstraction-level testing technique and offer insightful information for upcoming investigation and analysis. To fully understand how the methodology performs in various testing environments, more in-depth comparisons and tests will be made in subsequent case study results and analysis, to ensure the consistency, adaptability, and reliability of the methodology. Through these initiatives, we hope to increase the resilience and efficiency of the unified testing approach in a range of testing situations.

## 6. Conclusions

As demonstrated in this study, a unified and integrated testing methodology for spatially distributed wireless systems, in Industry 4.0 and Industrial IoT (IIoT) was proposed. This test methodology enables the use of the same test case description, execution, and visualization tool across multiple abstraction levels, including network simulation environments and physical testbed environments (in emulated testbed and field testbed environments). The significance of this infrastructure in comparing rival geographically dispersed wireless networking technologies—a critical need in IIoT applications such as remote monitoring, industrial automation, and smart manufacturing—was demonstrated through our NB-WWAN test campaign utilizing this implemented unified test system. This systematic examination and mapping of different test cases to the most suitable testing method form an essential part of our overall testing strategy, driving the effectiveness and efficiency of our NB-WWAN testing processes. By understanding the relationship between different testing methods and identifying the most suitable platforms for different test cases, we can ensure robust and reliable NB-WWAN network deployments. Future research should focus on improving the realism of network simulations and the flexibility of emulations, further enhancing their applicability across a broader range of test cases. Additionally, there is a need to refine our field-testing strategies to manage and control real-world variables better, ultimately increasing the reproducibility and reliability of the tests. By continuously refining our testing methodologies and adapting to technological advancements, we can ensure the seamless operation of NB-WWAN networks, laying the groundwork for a truly interconnected world. The authors utilize this unified testbed and testing methodology for many different test campaigns in different research projects; more performance assessment results are available and have been submitted for other publications [14]. Extension of this testbed will be carried out in the future to allow for smooth testing of time-synchronized wireless networks and 6G networks, further addressing the evolving needs of Industrial IoT.

## Figures and Tables

**Figure 1 sensors-24-07579-f001:**
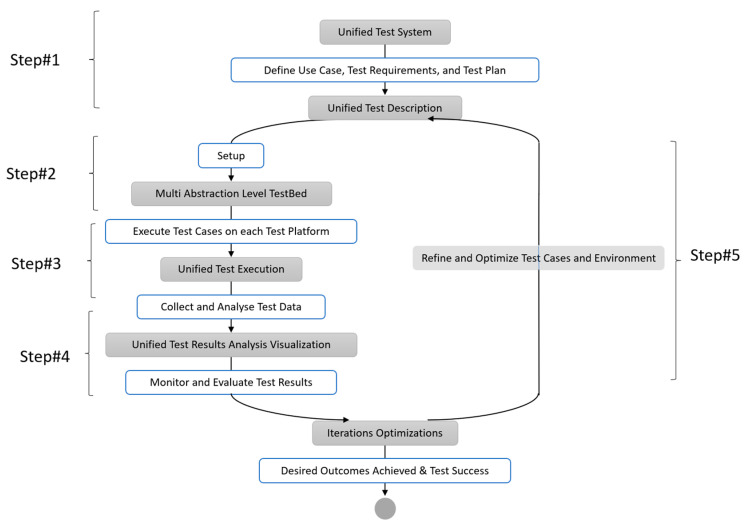
Unified multi-abstraction-level testing methodology.

**Figure 2 sensors-24-07579-f002:**
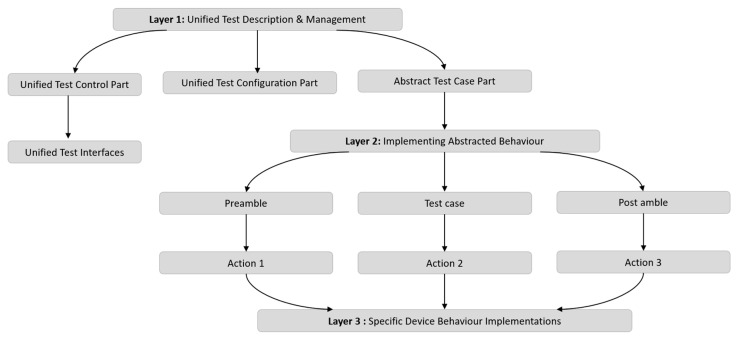
Concept of unified test description methodology.

**Figure 3 sensors-24-07579-f003:**
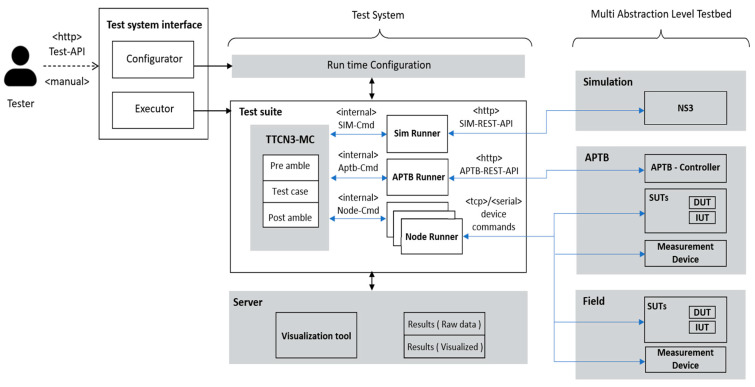
Unified test system architecture.

**Figure 4 sensors-24-07579-f004:**
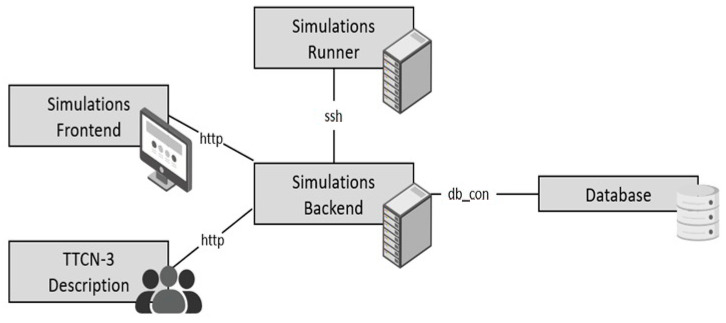
NS-3 architecture for integration with unified test system.

**Figure 5 sensors-24-07579-f005:**
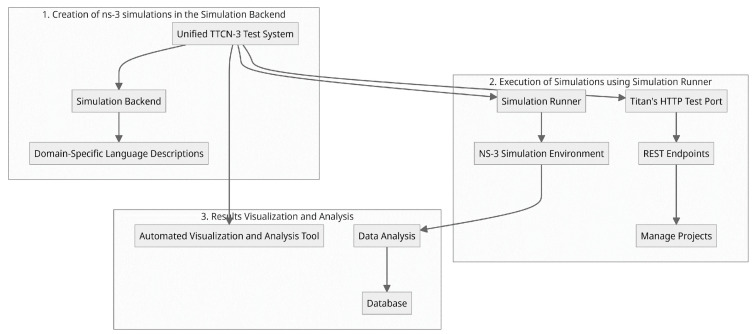
Steps in unified testing using integrated NS-3 to unified test system.

**Figure 6 sensors-24-07579-f006:**
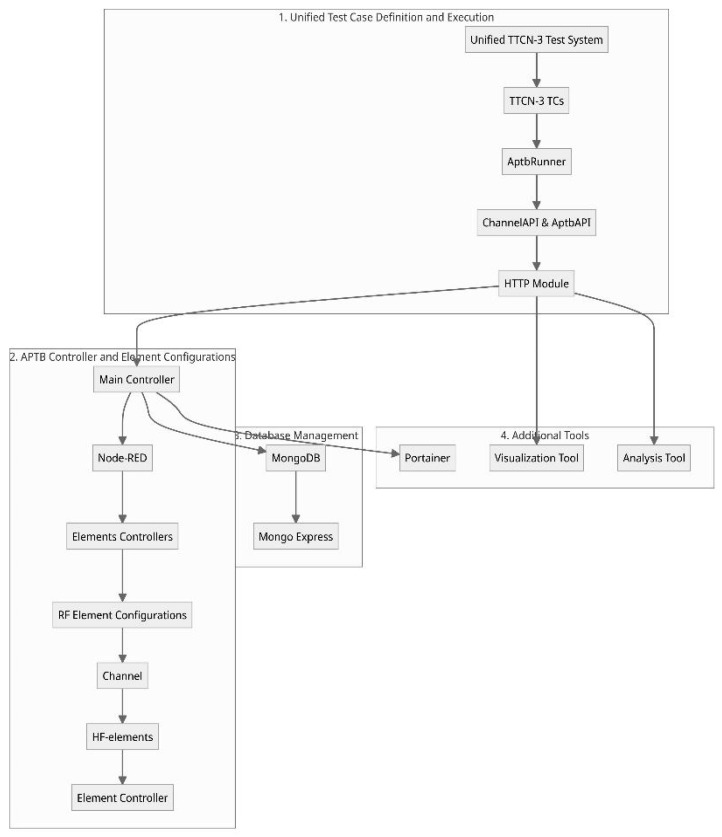
Architecture for integration of APTB to unified test system.

**Figure 7 sensors-24-07579-f007:**
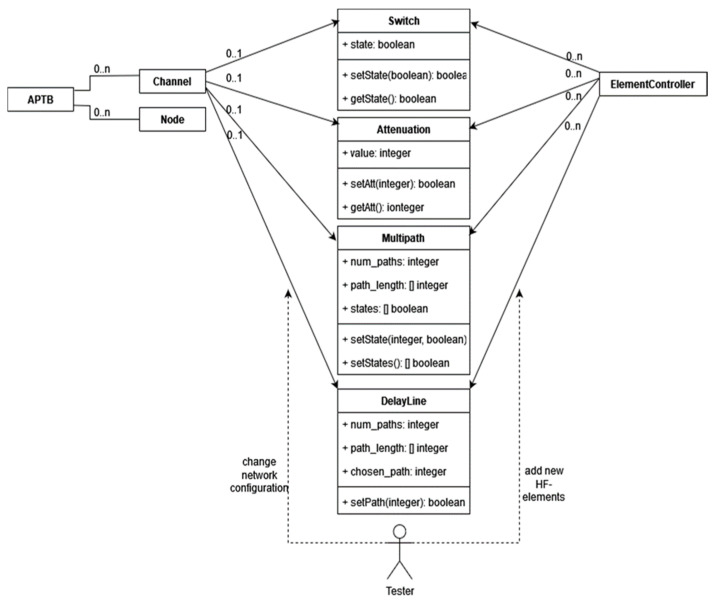
APTB channel characteristics and relations.

**Figure 8 sensors-24-07579-f008:**
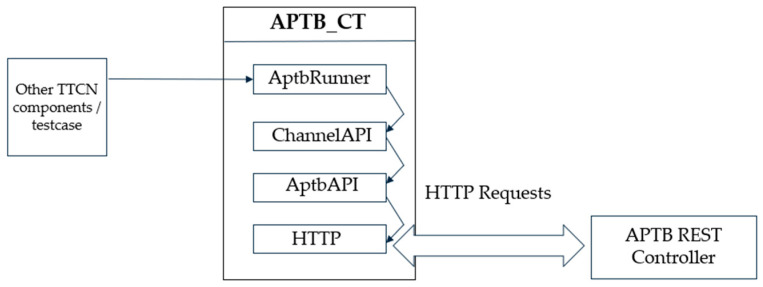
TTCN-3 module structure to control APTB.

**Figure 9 sensors-24-07579-f009:**
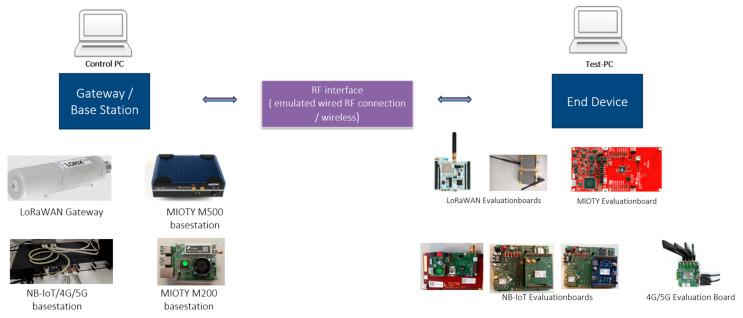
Various NB-WWAN Device Under Tests (DUTs) integrated into the test system.

**Figure 10 sensors-24-07579-f010:**
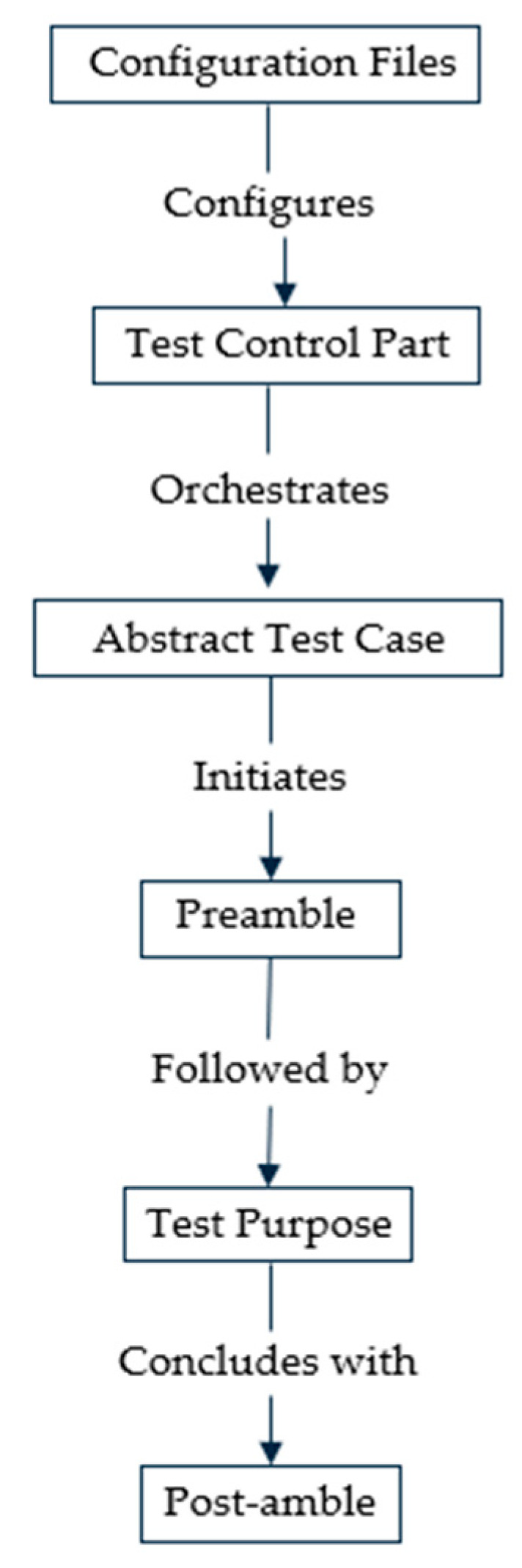
Unified test description procedures at Layer 1.

**Figure 11 sensors-24-07579-f011:**
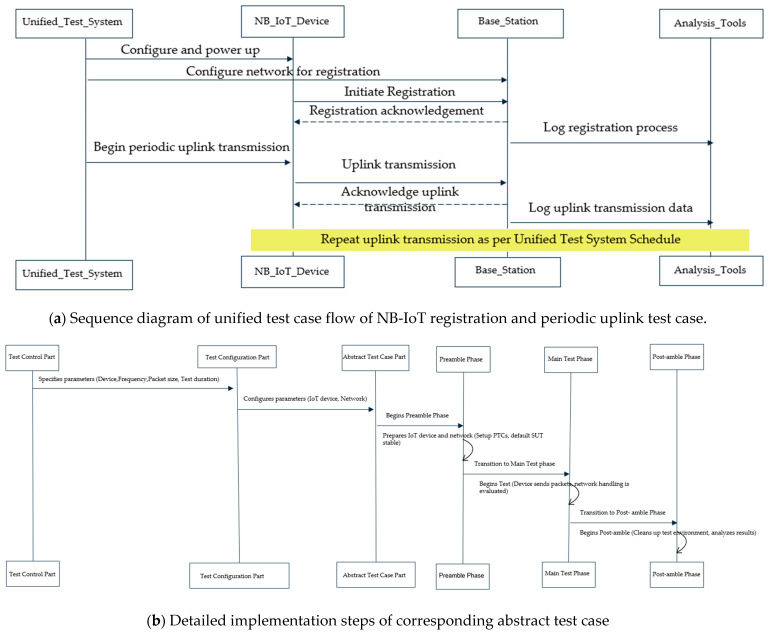
Detailed implementation concept of unified abstract test case example of periodic uplink.

**Figure 12 sensors-24-07579-f012:**
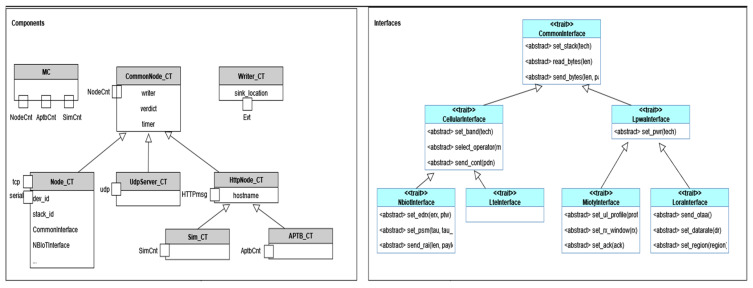
Example components and interfaces to define abstract behaviour.

**Figure 13 sensors-24-07579-f013:**
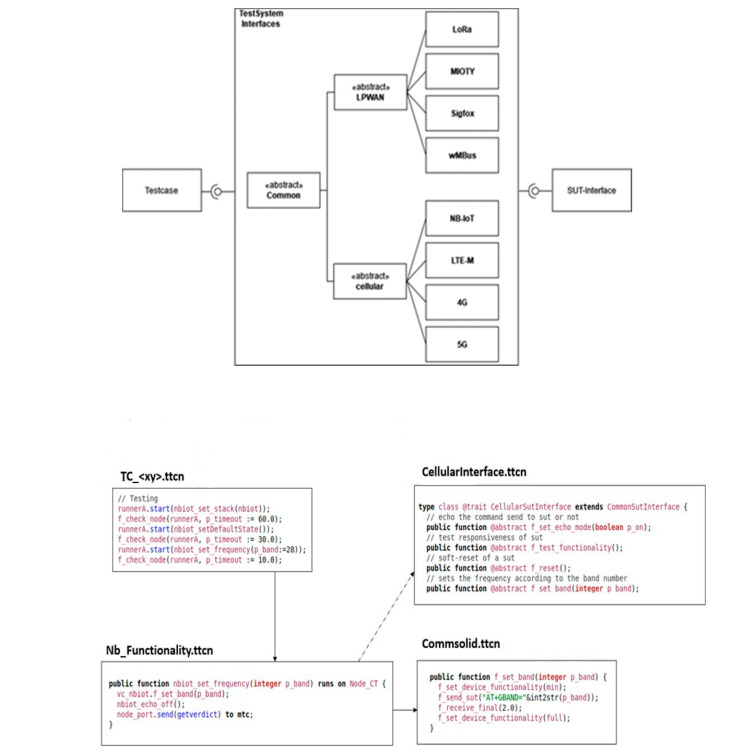
Unified test command mapping for an NB-IoT test case.

**Figure 14 sensors-24-07579-f014:**
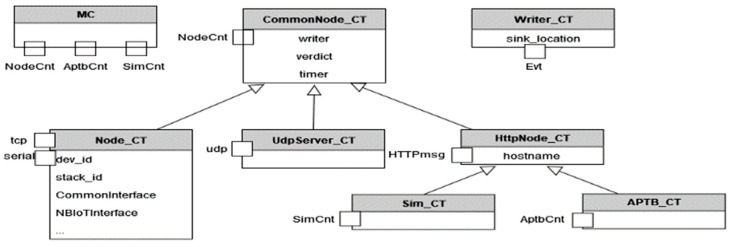
Mapping of components to port types.

**Figure 15 sensors-24-07579-f015:**
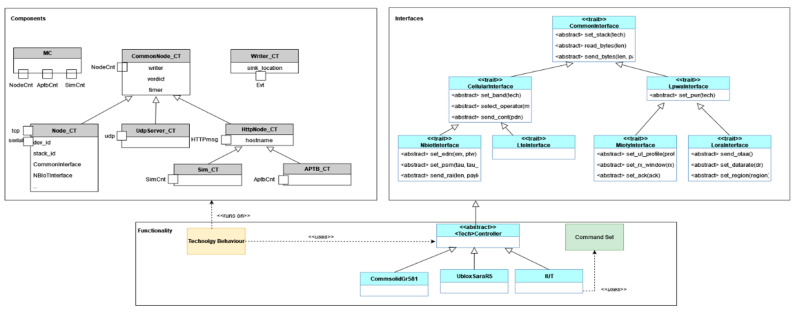
Concrete functionality implemented module example.

**Figure 16 sensors-24-07579-f016:**
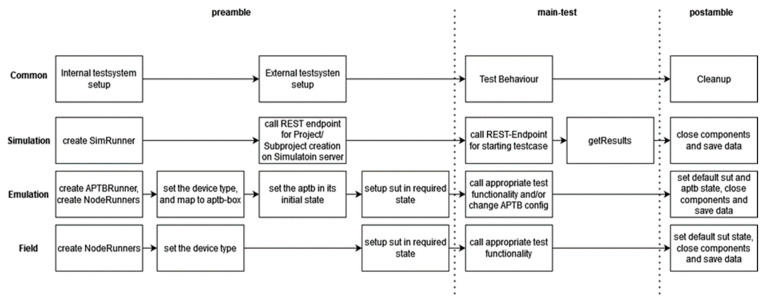
Unified test execution process in implemented test system.

**Figure 17 sensors-24-07579-f017:**
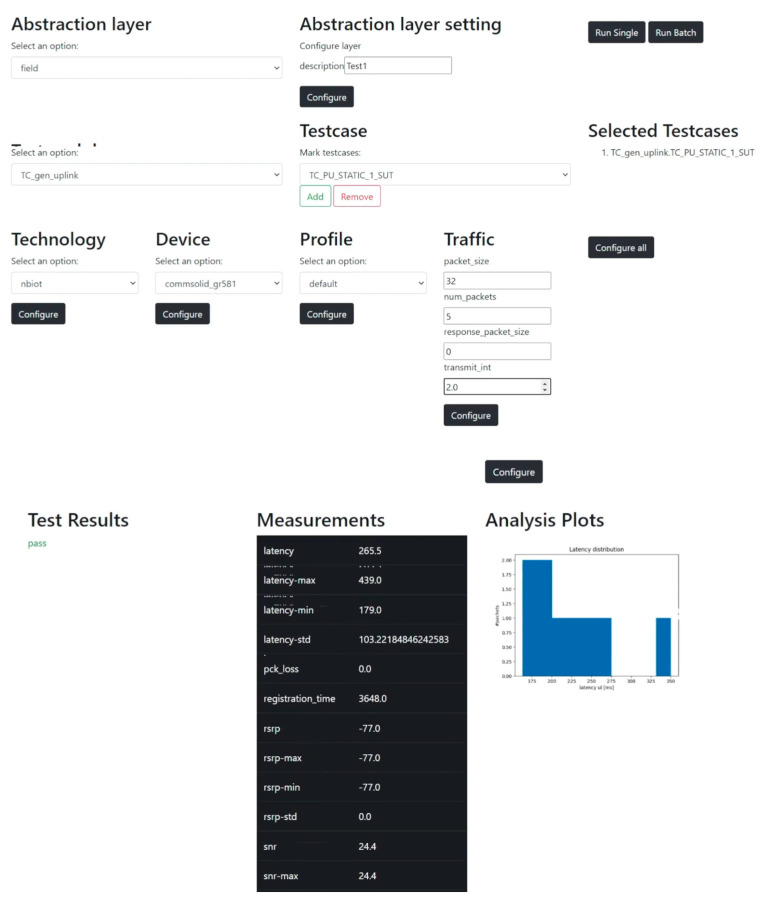
Unified test execution and visualization frontend of implemented test system.

**Figure 18 sensors-24-07579-f018:**
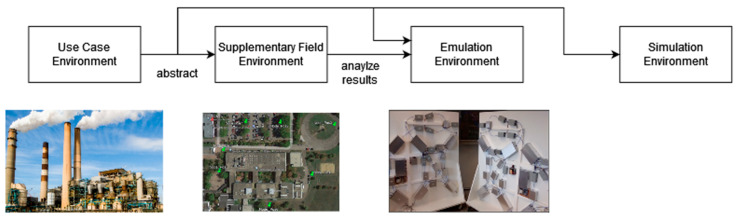
Unified test campaign environment.

**Figure 19 sensors-24-07579-f019:**
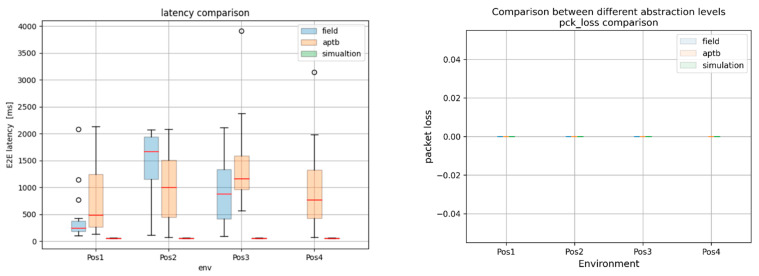
This figure represents a comparative analysis of KPIs: for an NB-IoT test across different levels of abstraction. The results indicate a significant deviation in performance, most notably in the simulation results, emphasizing the effects of abstraction.

**Figure 20 sensors-24-07579-f020:**
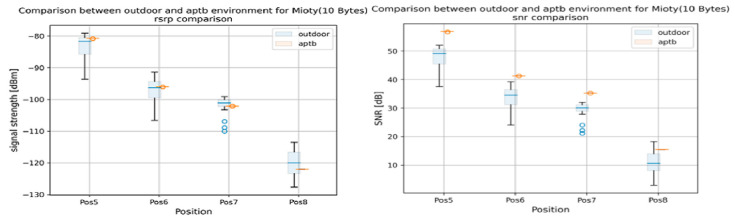
Comparison of RSRP and SNR in field and emulation environment.

## Data Availability

The source code, datasets, test results and prototype testbed generated and/or analysed during the study are not publicly available due to some of the information and implementation details being subject to a non-disclosure agreement with industrial partners of the project. Data are available from the corresponding author upon reasonable request.

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
