# Peer review of "Unified Multi-Abstraction-Level Functional Testing and Performance Measurements for Industrial IoT in Spatially Distributed Narrow Band-Wireless Wide Area Networks"

_sensors, 2024, doi:10.3390/s24237579_

Round 1

Reviewer 1 Report

Comments and Suggestions for Authors

The paper needs to be rewritten in terms of contents, structure, numbering, presentation, quality of figures, etc. The reviewer thinks the paper reads like a technical report, not a research paper. 

The paper needs:

- A better review of existing research

- A clear structure with numbered sections and good quality of figures

- More references to other research papers

- Clear results

The authors should rewrite the paper to be more concise and formal.

Comments on the Quality of English Language

There are several issues in terms of Quality of English Language. Careful language revision is needed in this paper.

1- Awkward phrasing: several sentences are structured awkwardly, making them difficult to understand. 

Examples:

"Towards 'Uniformity' in Functional Testing and Performance Measurements"

"uniformity defined as to the consistency and standardization"

"a thorough test strategy is required. The objectives, scope, technique, and particular test cases and scenarios"

"the interfaces between the test system and the system being tested are defined."

2-Grammatical Errors: there are many grammatical errors.

Examples:

"..protype implementation and its evaluation in this paper"

"..most of them are technology specific."

".because wireless communication devices frequently have limited resources.." 

"..testbeds attracted more and more attention.." 

"..as well as have a thorough understanding.."

"..an integrated and unified testbed for NB-WWANs is proposed, which enables the usage.."

Author Response

Reviewer-1 Comments- Response

Comment 1:

  • The paper needs to be rewritten in terms of contents, structure, numbering, presentation, quality of figures, etc. The reviewer thinks the paper reads like a technical report, not a research paper.
  • Response to Comment 1:
    • Thank you for detailed review of the paper.
    • We tried to make the paper as a research paper. In fact, in this paper, our aim is to present the testing methodology and then explain in detail the technical implementation details of the same in a recipe style. So that the readers and others, who wish to reproduce/ adapt our methodology for their specific set up. So more technical and implementation descriptions are given in the paper.
    • We tried rewrite technical contents more like research paper in different sections. For example, the 4.2.2 APTB Testbed Integration part is completely rewritten to read like research paper, as it had complex implementation details.
    • Structure and Numbering: We tried to restructure some parts and placed corresponding diagrams next to some complex concepts to make the structure and readability better. Also we avoided 4th level of numbering like 4.2.2.1 etc to simplify it.
    • Quality of the Figure: We tried to improve the quality of the figures by adjusting its resolutions and also by redrawing some (for e.g. Fig.8, 10,11 are redrawn).

Comment 2:

  • The paper needs:

- A better review of existing research

  • Response to Comment 2:
    • We have have conducted some more review of existing research and improved the section 2.2 with latest and related research work details. Two last paragraphs summarize review of existing research.

Comment 3:

  • A clear structure with numbered sections and good quality of figures
  • Response to Comment 3:
  • We tried to address the comment as much as possible. Numbering is simplified and consistent. And the quality of figured tried to improve/ redrawn.

Comment 4:

  • The authors should rewrite the paper to be more concise and formal.
  • We tried to address this as much as possible for now. As explained above we tried in this paper to provide a very detailed description of our implementations, we had previous publications in concise format or as an overview of the methodology or some of the results from our methodology, so our wish in this paper was to provide a very detailed insight into it.

Comment 5:

Quality of English Language

  • Response to Comment 5:
    • Thank you for your detailed feedback on the quality of the language in our paper. We have taken your comments seriously and have made some revisions to enhance readability, improve grammar. We will also aim for a editing of the paper by English experts to make them better, if paper accepts.
    • Below are the specific actions we have taken to address each concern.
    • "Towards 'Uniformity' in Functional Testing and Performance Measurements" was revised to “ Need of “Uniformity” in Functional Testing and Performance Measurements
    • "Uniformity defined as to the consistency and standardization" was rephrased to " uniformity refers to the consistency and standardization in the description and execution of tests across various System Under Test (SUT) types and at different levels of test abstraction”
    • The fragmented sentence "a thorough test strategy is required. The objectives, scope, technique, and particular test cases and scenarios" was restructured to "A comprehensive test strategy is required, encompassing clear objectives, scope, methodologies, and detailed test cases and scenarios."
    • "The interfaces between the test system and the system being tested are defined." was clarified as "clear definitions of interfaces between the test system and the system under test are established."
    • Grammatical errors mentioned were also tried to address
    • "..prototype implementation and its evaluation in this paper" was corrected toà the detailed technical description of the prototype implementation and its evaluation presented in this paper.
    • "..most of them are technology specific." was revised toà "..most of them are technology-specific."
    • ".because wireless communication devices frequently have limited resources.." -was modified to-> "because wireless communication devices often have limited resources."
    • "..testbeds attracted more and more attention.." was rephrased to-> "..testbeds have been attracting increasing attention."
    • "..as well as have a thorough understanding.." was revised to-> "..and to gain a thorough understanding."
    • "..an integrated and unified testbed for NB-WWANs is proposed, which enables the usage.." was clarified toà "we propose an integrated and unified testbed for NB-WWANs that enables the use of..."

We believe these revisions significantly enhance the paper's overall quality and readability and ensure alignment with academic standards.

Thank you again for your valuable feedback.

Reviewer 2 Report

Comments and Suggestions for Authors

The authors present unified multi-abstraction level testing methodology, protype implementation to evaluate different NB-WWAN technologies. This is an excellent practical paper. I think only the following comments need to be improved.

1. Authors should further introduce the implementation details of the automated physical testbed, such as product model and code source.

2. Authors should further explain why different campaign environments were chosen.

Comments on the Quality of English Language

Minor editing of English language required.

Author Response

Reviewer-2 Comments- Response

Comment 1: The authors present unified multi-abstraction level testing methodology, protype implementation to evaluate different NB-WWAN technologies. This is an excellent practical paper. I think only the following comments need to be improved.

  1. Authors should further introduce the implementation details of the automated physical testbed, such as product model and code source.

Response to Comment 1:

Thank you very much for the review and your comments about our paper and about its practical nature. We consider this paper as detailed implementation description paper of our testbed and methodology.

We have rewritten Automated Physical Testbed Part (4.2.2) to make the details clearer.

 The source code, data sets, test results and prototype testbed generated and/or analysed during the study are not publicly available due to that some of the information and implementation details are subject to non-disclosure agreement with industrial partners of the project. But are available from the corresponding author on reasonable request. Our team also have a spin-off commercial company and transfer model with the testbed product. Also within our university transfer offering automated physical testbed product is also make available for externals for evaluations. There were many usage by other researchers and companies to those as well. 

Comment 2. Authors should further explain why different campaign environments were chosen.

Response to Comment 2:

We tried to explicitly address this point on the Pefromance evaluation part. In the paper we propose a unified test methodology applicable for the different abstraction levels ( such as simulation, emulation and field tests). So to determine the methodology's consistency and efficacy across various abstraction levels, it was tested in three separate environments: simulation, emulation, and field testing was chosen as the test campaign environment.

Comment 3:

Minor editing of English language required.

Response to Comment 3: We also tried to correct the errors we came across.

Thank a lot once again for your kind feedback, we tried to address them our level best to improve the overall quality of the paper.

Reviewer 3 Report

Comments and Suggestions for Authors

The paper presents a testing methodology for wireless systems, with a focus on low-power distributed networks for IoT. A five-step methodology is introduced, which is highly detailed, and the paper concludes with an application of this methodology in a real-world scenario, including both simulation and field testing.

In my view, the topic is interesting, and the proposals have merit, as they are based on software testing concepts and wireless systems. 

However, I believe that both the topic and the depth of analysis may not be suitable for the journal. When proposing a methodology, it is essential to include a comparison with other approaches in the literature. This section of the article needs further development. Perhaps a table in Section II with the closest related works, highlighting key comparison points, would be beneficial.

Additionally, the introduction should clearly state the main contributions of the paper in a more direct and objective manner, providing the reader with a clear indication of what will be addressed in the remainder of the work.

I also believe the methodology should aim to simplify the professional’s workflow, but the current proposal is too complex. Simplifying certain steps might make it more appealing.

Comments on the Quality of English Language

English is ok.

Author Response

Reviewer-3 Comments- Response

Comment 1:  The paper presents a testing methodology for wireless systems, with a focus on low-power distributed networks for IoT. A five-step methodology is introduced, which is highly detailed, and the paper concludes with an application of this methodology in a real-world scenario, including both simulation and field testing.

In my view, the topic is interesting, and the proposals have merit, as they are based on software testing concepts and wireless systems.

Response to Comment 1:

Thank you very much for the review and your comments about our paper and about approach of combining testing concepts to wireless systems. We consider this paper as detailed implementation description paper of our testbed and methodology.

Comment 2. However, I believe that both the topic and the depth of analysis may not be suitable for the journal. When proposing a methodology, it is essential to include a comparison with other approaches in the literature. This section of the article needs further development. Perhaps a table in Section II with the closest related works, highlighting key comparison points, would be beneficial.

Response to Comment 2: We tried to address the point mentioned. We have further conducted literature review and summarized more related works. Section 2.2 rewritten as “Analysis of Related Works of Functional Testing and Performance Measurement Methodologies of NB-WWAN Systems”. The last two paragraphs are new and summarized a detailed analysis of related works

Comment 3: Additionally, the introduction should clearly state the main contributions of the paper in a more direct and objective manner, providing the reader with a clear indication of what will be addressed in the remainder of the work.

Response to Comment 3:  to address the same, we added main contributions at the end of introduction section as below.

--

In this paper, we present a unified methodology for functional testing and performance measurements, and its prototype implementation specifically designed to address the challenges of diverse System Under Test (SUT) technologies and devices across varying test abstraction levels. Our approach allows for high-level test case descriptions that can be consistently applied across simulation, emulated testbeds, and real field environments. Additionally, we share insights from initial performance evaluations, demonstrating the methodology’s effectiveness in achieving standardized testing and comparability across platforms. This work contributes a unified and systematic testing methodology that promotes consistency, repeatability, and scalability in functional testing and performance measurement methodology for future wireless IoT systems.

Thank a lot once again for your kind feedback, we tried to address them our level best to improve the overall quality of the paper.

Round 2

Reviewer 1 Report

Comments and Suggestions for Authors

The revised paper incorporates many suggested changes and now somehow addresses the review comments. 

Comments on the Quality of English Language

The English is good, but further refinement is necessary to achieve more clarity and fluency. 

Author Response

The reviewer #1 had provided their suggestions on the English
Improvement in this revision request.  In this minor revision these were addressed in detail throughout the paper and submitting the tracked version for reference. Thank you very much for valuable review.

Reviewer 3 Report

Comments and Suggestions for Authors

The changes were made in alignment with the requested specifications.

Author Response

 In this minor revision round, I tried to improve the quality of English language in detail throughout the paper and submitting the tracked version for quick reference. Thank you very much for valuable review.